# Multiple-P450 Gene Co-Up-Regulation in the Development of Permethrin Resistance in the House Fly, *Musca domestica*

**DOI:** 10.3390/ijms24043170

**Published:** 2023-02-06

**Authors:** Ming Li, Xuechun Feng, William R. Reid, Fang Tang, Nannan Liu

**Affiliations:** 1Department of Entomology and Plant Pathology, Auburn University, Auburn, AL 36849, USA; 2Department of Entomology, University of California, San Diego, CA 92093, USA; 3Institute of Infectious Diseases, Shenzhen Bay Laboratory, Shenzhen 518000, China; 4Department of Biochemistry, State University of New York at Buffalo, Buffalo, NY 14203, USA; 5College of Forestry, Nanjing Forestry University, Nanjing 210037, China

**Keywords:** insecticide resistance, P450 gene overexpression, gene functional characterization, *D. melanogaster* transformation, in vitro metabolism, homology modeling

## Abstract

This paper reports a study conducted at the whole transcriptome level to characterize the P450 genes involved in the development of pyrethroid resistance, utilizing expression profile analyses of 86 cytochrome P450 genes in house fly strains with different levels of resistance to pyrethroids/permethrin. Interactions among the up-regulated P450 genes and possible regulatory factors in different autosomes were examined in house fly lines with different combinations of autosomes from a resistant house fly strain, ALHF. Eleven P450 genes that were significantly up-regulated, with levels > 2-fold those in the resistant ALHF house flies, were in CYP families 4 and 6 and located on autosomes 1, 3 and 5. The expression of these P450 genes was regulated by trans- and/or cis-acting factors, especially on autosomes 1 and 2. An in vivo functional study indicated that the up-regulated P450 genes also conferred permethrin resistance in *Drosophila melanogaster* transgenic lines. An in vitro functional study confirmed that the up-regulated P450 genes are able to metabolize not only cis- and trans-permethrin, but also two metabolites of permethrin, PBalc and PBald. In silico homology modeling and the molecular docking methodology further support the metabolic capacity of these P450s for permethrin and substrates. Taken together, the findings of this study highlight the important function of multi-up-regulated P450 genes in the development of insecticide resistance in house flies.

## 1. Introduction

Efforts to characterize the molecular mechanisms involved in the development of insecticide resistance in insect pests have generally focused on building a better understanding of the mechanisms involved, providing vital information for practical applications such as the design of novel strategies to prevent or minimize the spread and evolution of resistance development and the control of insect pests [1]. The interactions among multiple mechanisms or genes in response to the development of insecticide resistance have been extensively explored; moreover, the transcriptional up-regulation of the detoxification machinery responsible for increasing the metabolism of insecticides into less harmful substances and facilitating insecticide excretion is known to play an important role in allowing insects to defend themselves against insecticides [2,3]. Cytochrome P450s are known to be important components in the detoxification machinery of insects; their transcriptional up-regulation in insects increases both protein production and enzymatic activities, which, in turn, enhance the metabolic detoxification of insecticides and plant toxins, thus leading to the development of insecticide resistance [1,4]. Multiple P450s have also been shown to work in combination to boost insecticide resistance through co-up-regulation in resistant insects [1,5,6,7,8,9,10]. Recent advances in genome/whole transcriptome sequencing technology represent a major milestone, opening up new ways for researchers to explore precisely how many P450 genes are involved in insecticide resistance in individual insect pests [11,12,13,14,15,16,17]. 

The house fly, *Musca domestica*, is a major domestic, medical and veterinary pest that can transmit over 200 human and animal pathogens [18]. Control efforts have mainly relied on the application of insecticides, primarily pyrethroids [19], but the major barrier hampering house fly control, as in other insect species, is their remarkable ability to develop not only resistance to the insecticide used against them but also cross-resistance to unrelated classes of insecticides [4,12]. The house fly is widely used as a useful model with which to study and predict insecticide resistance because of its ability to rapidly develop resistance and cross-resistance to insecticides, its well-established linkage map for five autosomes and two sex chromosomes (X and Y), its relatively well-understood biochemistry and genetics of insecticide resistance, and its readily available genomic and transcriptome data analyses [11,12]. Building on our earlier adult house fly transcriptome data analysis [11], where cytochrome P450s were found to be the major functional group up-regulated in resistant house flies, the current study moved on to characterize the total number of P450 genes directly involved in resistance in a single resistant insect. A systematic study conducted at the whole transcriptome level was used to develop an in-depth understanding of these P450 genes, with a focus in how their co-up-regulation, interaction, metabolism and function contribute to the development of insecticide resistance in house flies.

## 2. Results

### 2.1. Cytochrome P450 Genes and Their Transcriptome Expression Profiles in M. domestica

There are 146 P450 genes identified in the genome database of house flies [12], of which 86 are expressed in ALHF adult house flies based on our previous whole transcriptome analysis [11]. These cytochrome P450s were selected to pinpoint the most important P450s involved in the development of permethrin resistance for this study. The majority of these 86 P450 genes were assembled into clans 3 and 4, with thirty-nine P450 genes in clan 3, of which twenty-seven were in family 6, six were in family 9, three were in family 28, one was in family 310 and one was in family 317 (Table 1 and Appendix A). Of the thirty P450 genes in clan 4, twenty-three were in family 4, two were in family 313, one was in family 311 and one was in family 318. Of the remaining P450 genes, four were in families 304, 305, 306 and 18 of clan 2, and thirteen P450 genes were found to be in five families in the mitochondrial clan, namely 12, 301, 302, 314 and 315. Interestingly, the detectable P450 genes in the ALHF house flies showed a clear expansion in families 4, 6 and 9 compared to the other families. P450 genes in these three families have previously been implicated in environmental response, xenobiotic metabolism and the development of resistance in insects [20,21,22,23]. 

The relative expression profile of the 86 P450 genes was examined in resistant ALHF and two susceptible aabys and CS house fly strains to eliminate false positive results. The expression levels of 11 P450 genes were found to be significantly (*p* < 0.05) up-regulated, with levels > 2-fold higher in the ALHF house flies compared to the levels in both the aabys and CS house flies (Table 2 and Appendix A). These genes were distributed into two clans (clans 3 and 4), with five genes in family 4 and six in family 6. Fourteen P450 genes were found to be significantly down-regulated in the ALHF strain compared to both the aabys and CS strains (Appendix A). The remaining 59 P450 genes showed no significant difference in expression levels between ALHF and either or both the aabys and CS strains.

### 2.2. Autosome Location and Co-Up-Regulation of P450 Gene Expression in Insecticide-Resistant House Flies

Autosomal location analyses were conducted for all 11 of the up-regulated P450 genes using allele-specific PCR (AS-PCR) determination utilizing five house fly BC_1_ lines, and allele-specific primer pairs designed based on the specific sequence of the genes from ALHF, by placing a specific nucleotide polymorphism at the 3′ end of one primer of each primer pair to facilitate the preferential amplification of specific alleles from ALHF [24]. Our results show that the ALHF allele-specific primer sets for *CYP6D3* and *CYP6D10* amplified specific DNA fragments only in flies containing the autosome 1 wild-type marker from ALHF (Figure 1), confirming that these two P450 genes are indeed on autosome 1. Similarly, *CYP4G13*, *CYP4G99* and *CYP4S24* were found to be located on autosome 3, and *CYP4E10*, *CYP4E11*, *CYP6A36*, *CYP6A40, CYP6A52* and *CYP6A58* on autosome 5 (Figure 1). 

To better understand the cis-/trans-regulation of the up-regulated P450 genes in ALHF house flies, we next determined the effects and co-regulation of various factors on the expression of the up-regulated P450 genes by analyzing the gene expression changes resulting from autosome replacement in ALHF. This enabled us to evaluate the role that the factors on each autosome plays in P450 gene overexpression in ALHF. The P450 gene expression levels were examined in resistant ALHF, susceptible aabys and five house fly homozygous lines: A2345, A1345, A1245, A1235 and A1234; these lines represent the ALHF strains where autosomes 1, 2, 3, 4 and 5, respectively, have been replaced by the autosome from aabys. We found no significant change in the level of expression in any of the P450 genes tested when autosome 4 of ALHF was replaced with that from aabys (i.e., line A1235; Figure 2, Table 2), strongly suggesting that the factors/genes on autosome 4 do not play a major role in the up-regulation of P450 genes in ALHF. This agrees with the findings reported in previous studies by Tian et al. [25], Liu and Scott [26], and Liu and Yue [27]. 

Clear changes in the P450 gene expression levels were observed when any one of autosomes 1, 2, 3 and 5 in ALHF was replaced by its aabys counterpart. Apart from one P450 gene, *CYP6A52*, whose up-regulation was solely controlled by *cis* factor(s) on autosome 5, where the P450 gene itself is located, the expression of the remaining 10 up-regulated P450 genes were all linked to factors on more than one autosome (Figure 2, Table 2). These results suggest that *cis*/trans factors are capable of co-regulating the P450 gene expression responsible for insecticide resistance. Of the five P450 genes (*CYP4E10*, *CYP4E11*, *CYP6A36*, *CYP6A40* and *CYP6A58*) located on autosome 5, the expression of three (*CYP4E10*, *CYP6A36*, and *CYP6A58*) was controlled by trans-regulatory factors on autosome 1 and 2, while that of the other two (*CYP4E11* and *CYP6A40*) was controlled by trans-regulatory factors on autosome 2 only. Of the three genes located on autosome 3, two (*CYP4G13* and *CYP4S24*) were found to be regulated by the trans-regulatory factors on autosomes 1 and 2, and the other (*CYP4G99*) by trans-regulatory factors on autosome 1. One of the two genes located on autosome 1 (*CYP6D10*,) was up-regulated by trans-regulatory factors on autosomes 2 and 5, while the other (*CYP6D3*) was regulated by trans-regulatory factors on autosome 2 (Figure 2, Table 2). Clearly, factors on autosomes 2 and 5, especially those on autosome 2, are commonly involved in the up-regulation of P450 gene expression associated with the development of resistance in house flies. 

### 2.3. Up-Regulated P450 Genes in Transgenic D. melanogaster—An Alternative/Independent Approach to Defining P450 Gene Functions

To further validate the function of the up-regulated P450 genes in the development of insecticide resistance in ALHF house flies, three P450 genes, *CYP4S24*, *CYP6A36*, and *CYP6D10*, were selected for a transgenic study using the GAL4-UAS enhancer trap system of *D. melanogaster*. These three genes represent the up-regulated P450 genes on each of three different autosomes: the expression of *CYP4S24*, located on autosome 3, is regulated by trans-regulatory factors on autosomes 1 and 2; the expression of *CYP6A36*, located on autosome 5, is regulated by trans-regulatory factors on autosomes 1 and 2; and the expression of *CYP6D10*, located on autosome 1, is regulated by trans-regulatory factors on autosomes 2 and 5. 

The first step was to confirm the presence of house fly P450 genes in the transgenic lines of *D. melanogaster* via RT-PCR; the results show that all the transgenic lines did indeed express the target transgenes (Figure 3A). To test whether the GAL4-UAS expression system in *D. melanogaster* increases the expression of target house fly P450 genes, qRT-PCR was further employed to detect the expression levels among crosses of the transgenic lines including: a control (Bloomington stock #24484 *D. melanogaster* line containing an empty pUAST vector), GAL4 (the ubiquitous Act5C driver line), the control crossed with GAL4 (the F1 progeny produced by a cross between the control females and the GAL4 males), the two homozygous transgenic P450 lines, and the P450 transgenic lines crossed with the GAL4 driver line (the F1 progeny from a cross between the P450 homozygous transgene line females and the GAL4 males). There was no sign of the expression of any of the three P450 genes in the non-transgenic *D. melanogaster* lines (control, GAL4 and control × GAL4), but it was detected in both of the transgenic lines (P450 and P450 × GAL4). The expression of these P450 genes was enhanced by the GAL4-UAS expression system of *D. melanogaster* (P450 × GAL4 > P450) (Figure 3B).

The next step was to characterize the sensitivity of the non-transgenic and transgenic *D. melanogaster* lines to permethrin. The bioassay results showed no significant difference in permethrin toxicity among the three non-transgenic lines based on overlapping 95% confidence intervals. However, transgenic *D. melanogaster* lines with house fly P450s showed significantly increased levels of resistance to permethrin compared to non-transgenic lines (Figure 3C). Of the three P450s tested, transgenic flies with the house fly *CYP6A36* gene had the highest level of resistance to permethrin (a ~6.5-fold increase compared with the control lines), followed by flies with *CYP6D10* (a ~4-fold increase) and *CYP4S24* (a ~2-fold increase). These results indicate that these up-regulated P450 genes in the house fly boosted permethrin resistance in *D. melanogaster*, highlighting the important function of these P450 genes in the permethrin resistance of *M. domestica*. 

### 2.4. Functional Expression of P450s and In Vitro Metabolism Studies

Functional validation to confirm that metabolic enzymes encoded by the overexpressed metabolic genes do indeed metabolize the insecticides in resistant mosquitoes is crucial in efforts to pin down the gene’s precise function in the metabolism of insecticides and in the development of insecticide resistance. We therefore heterogeneously expressed CYP4S24, CYP6A36 and CYP6D10 proteins with house fly cytochrome P450 reductase (CPR) in a baculovirus expression system. Each of the P450s was co-expressed with CPR via simultaneous infection of *Sf*9 cells through two recombinant viruses (P450-recombinant baculovirus (P450rbv) with an MOI of 0.5 and CPR-recombinant baculovirus (CPRrbv) with an MOI of 0.05) in a standard suspension culture [28]. The functional catalytic activity of these recombinant P450/CPR proteins was confirmed via an ECOD assay, where 7-ethoxycoumarin was hydrolyzed to 7-hydroxycoumarin by recombinant P450 and CPR proteins with different rates of metabolism (Appendix A). CYP6D10/CPR exhibited the highest ECOD activity, followed by CYP6A36, and then, CYP4S24, confirming that all three of these recombinant P450/CPR proteins can indeed convert significant amounts of the substrate 7-ethoxycoumarin to the fluorescent product 7-hydroxycoumarin. No significant P450 activity was observed in proteins isolated from the original *Sf*9 cells.

Permethrin metabolic activity was assayed using microsomal proteins of P450/CPR in the presence or absence of NADPH. The degradation of the substrate was monitored via reversed-phase HPLC. Our in vitro study demonstrated that all three P450s and CPR co-expressed microsomal proteins and metabolized both trans- and cis- permethrin. Significant levels of the permethrin were metabolized by the three P450s and CPR co-expressed microsomal proteins in a 20 µM permethrin solution containing both cis- and trans-isomers compared to a no-NADPH control after a 120 min incubation period. Of the three P450s, CYP6A36 showed the highest activity toward both trans- and cis-permethrin (64.27% and 59.81% of the trans- and cis-permethrin, respectively, was eliminated in 120 min), closely followed by CYP6D10 (50.55% and 43.01% of the trans- and cis-permethrin, respectively). Meanwhile, CYP4S24 exhibited the lowest activity toward trans- and cis-permethrin (37.53% and 30.61% of the trans- and cis-permethrin, respectively) (Figure 4). These results clearly demonstrate that all three up-regulated P450s are capable of metabolizing permethrin in vitro.

It has been reported that pyrethroids can be hydrolyzed by carboxylesterases to produce PBalc and PBald, and these metabolites can be further processed to 4′-OH PBalc and PBald by P450s in mammals and mosquitos [29,30,31,32,33,34,35]. We therefore conducted a metabolism assay of PBalc and PBald using house fly P450s and CPR co-expressed microsomal proteins. All three P450s metabolized both PBalc and PBald, with CYP4S24 exhibiting the highest ability to metabolize both PBalc and PBald (61.20% and 92.97% of the PBalc and PBald, respectively, was eliminated in 120 min) (Figure 4). Our results confirm that CYP6A36, CYP6D10 and CYP4S24 are not only able to metabolize cis- and trans-permethrin, but can also metabolize permethrin’s metabolites, PBalc and PBald.

### 2.5. Homology Modeling and Substrate Docking

To understand the underlying metabolic mechanisms and the differences between the responses of these selected P450s toward permethrin, PBalc and PBald, homology modeling and molecular docking simulations were conducted. Not surprisingly, these P450s were found to contain a number of conserved P450 characteristics, including the oxygen binding motif of helix I (A/GGxE/DTT/S), the heme-binding signature (FXXGXRXCXG) near the C-terminal, the EXXR motif which stabilizes the heme core, and six predicted substrate recognition sites (SRSs) (Appendix A). 

The active site for P450 is buried deep within the enzyme structure. It is connected to the surrounding environment by a network of channels that serve as access/egress paths and may determine the substrate specificity of P450s [36,37]. The most probable candidate substrate access/egress channels (solvent and family 2) were therefore investigated in this study. An analysis of the geometry involved showed that eight channels (2a, 2ac, 2b, 2c, 2d, 2e, 2f and S) were present in CYP6A36 and CYP6D10 (Figure 5). These channels merge together to form a large opening to the heme prosthetic group, and thus, provide a large active cavity volume. The seven channels (2a, 2ac, 2b, 2c, 2d, 2f and S) detected in CYP4S24 formed a much more restricted narrow opening to the heme prosthetic group and a small active cavity, likely due to two amino acids (Glutamate 313 and Threonine 317) that protrude into the helix I structure (Figure 5).

A comparison of permethrin, PBalc and PBald binding conformations at the active sites of the P450 models revealed several possible binding and metabolic mechanisms of P450s to these chemicals. The predicted active site of these three P450s easily accommodates both cis- and trans-permethrin. Although permethrin can dock with the gem-dimethyl group, with 5-phenoxybenzyl carbon and 4′-phenoxybenzyl carbon closer to the heme center of the active sites of CYP6A36 and CYP6D10, docking simulations suggest that 4′-hydroxylation is a major route for metabolism (Figure 6A–D, Appendix A), which agrees with the findings reported in other studies [29,38,39,40,41]. Only one binding conformation has the gem-dimethyl groups of the cyclopropane moiety approaching the heme, which, when modeled, produces cis- or trans-hydroxymethyl-permethrin at the active site of CYP4s24 (Figure 6E,F, Appendix A). This has previously been identified as a minor route of pyrethroid metabolism [38]. 

The high binding affinity and the short distance from the substrate metabolic sites to the P450 heme iron are responsible for the high electrophilic character of this position [42] and the high catalytic reactivity [40]. Interestingly, our docking study not only showed that the catalytic reactivity of these three P450s is potentially greater with trans-permethrin compared to cis-permethrin, as trans-permethrin docked closer to the heme center and presented a higher binding affinity, but also demonstrated that CYP6A36 has the strongest ability to metabolize permethrin compared with CYP4S24, which had the lowest ability to metabolize permethrin (Appendix A). These findings are consistent with those of our transgenic *D. melanogaster* study and in vitro permethrin metabolism study.

In CYP6A36 and CYP6D10, PBalc and PBald docked with productive poses such that the 4′-phenoxybenzyl carbon was oriented above the heme (Figure 7A,B,D,E, Appendix A), creating two products (4′-hydroxylated PBalc (4′-OH-PBalc) [34] and 4′-hydroxylated PBald (4′-OH-PBald), respectively), as predicted. However, in CYP4S24, PBalc and PBald docked with the carbon atom in the hydroxyl and the aldehyde group directly over the heme, with high binding affinity and only a short distance from the heme iron (Figure 7C,F, Appendix A). This positioning may lead to the production of 3-phenoxybenzoic acid (PBAcid), which has been identified as the final metabolite of pyrethroids [29,31,34,35].

## 3. Discussion

Previous studies have suggested that the constitutively increased expression and induction of P450 genes in insects is responsible for increased levels of total P450 and P450 activities, which significantly affect the disposition of xenobiotics or endogenous compounds in the tissues of organisms, and thus, alter their toxicological effects. Multiple P450s are commonly thought to share roles associated with the development of insecticide resistance through co-regulation in resistant insects [7,8,24,43,44,45,46]. Using coupled CRISPR-Cas9-based genome editing techniques and in vitro metabolism analysis, Wang et al. [47] approved a cluster of nine P450 genes involved in the metabolism of xenobiotics in cotton bollworms, *Helicoverpa armigera*. While many studies have revealed that the overexpression of P450 genes is regulated by unknown trans- or cis-regulatory factors [1,11,26,27,43,44,48,49,50,51,52,53,54], mutations in P450 proteins have also been proven to impact the xanthotoxin metabolism of P450s [55]. Taken together, these studies have demonstrated not only that insecticide resistance in a single insect is conferred by the co-contribution of multiple P450 genes, but also that the interaction with regulatory factors plays an important role. In view of the above, the current paper presents a systematic study conducted at the whole transcriptome level to characterize the P450 genes known to be involved in the development of pyrethroid resistance, by analyzing the expression profiles of a total of 86 cytochrome P450 genes [11] in house fly strains with different levels of resistance to pyrethroid/permethrin. A total of 11 P450 genes that were found to be significantly up-regulated with levels > 2-fold those in resistant ALHF house flies were in families 4 and 6, located on autosomes 1, 3 and 5. Except for *CYP6A52*, whose up-regulation was solely controlled by the cis factor(s) on a single autosome, the up-regulated P450 genes were shown to be up-regulated through interactions with factors on more than one autosome (primarily autosomes 1 and 2) suggesting that the expression of up-regulated P450 genes is controlled by the cis or trans factors in resistant house flies. Hence, further characterizing the regulatory factors that are involved in resistance P450 gene expression will not only support efforts to elucidate the mechanisms of insecticide resistance development, but also facilitate work being carried out to diagram the entire transcriptional regulatory network and select new, more efficient targets for insect pest control.

While our knowledge regarding the regulation of insect P450 gene expression in insecticide resistance and the identification of regulatory factors in insect P450 expression is still far from complete for most insect P450s, several recent studies have clearly demonstrated the ambiguity of the transcriptional regulation mechanisms involved in the regulation of resistance P450 gene expression through cis elements and trans-regulatory factors [2,56,57,58,59,60,61,62,63]. A few of the regulatory factors known to be involved in insect P450 gene expression in insecticide resistance, including orphan nuclear receptors, cap ‘n’ collar C (*CncC*)/muscle aponeurosis fibromatosis (*Maf*), G-protein-coupled receptors (GPCRs), and the cAMP-response element binding protein, have been identified in several insect species [54,57,59,60,64]. Hu et al. [65] have reported the constitutive up-regulation of transcription factors CncC/Maf to be at least partially responsible for the upregulated expression of CYP321A8 in a resistant strain of *Spodoptera exigua* and demonstrated that CncC/Maf enhanced the expression of CYP321A8 by binding to specific sites in the promoter. The authors also found that cis-regulatory elements resulting from a mutation in the CYP321A8 promoter in a resistant strain facilitate the binding of the orphan nuclear receptor, Knirps, and enhance promoter activity. Yang et al. [64] discovered that the overexpression of *CYP6CM1* in white flies, known to be responsible for the development of resistance to neonicotinoid insecticides, is regulated via the trans-regulatory factor cAMP-responsive element modulator (CREM), the activation of which is mediated by the mitogen-activated protein kinase (MAPK) signaling pathway. A GPCR regulatory pathway through Gαs, AC, PKA and cAMP in mosquitoes that has been implicated in the regulation of multiple resistance P450 gene expression levels, eventually leading to the enhanced detoxification of insecticides in the mosquito *Culex quinquefasciatus*, has been systematically studied and modeled [2,54,58,59]. The importance of GPCR genes and GPCR signaling pathways in the regulation of resistance P450 genes has also been reported in other resistant insect species, including *Lymantria dispar* [66], *Culex pipiens* [67] and *M. domestica* [68]. Li et al. [2,54] conducted functional studies of rhodopsin-like GPCR using transgenic lines of *D. melanogaster* and found that not only did the tolerance to permethrin insecticide increase in the transgenic lines of *D. melanogaster*, but the expression of *Drosophila*’s resistance the P450 361 genes *CYP12d1 and CYP6a8* also increased. Similar functional studies using *Drosophila* transgenic lines have further confirmed the involvement of GPCRs in the regulation of P450 gene expression and insecticide resistance in *L. dispar* [66] and *M. domestica* [68]. Transposable elements (TEs), DNA sequences with the ability to change their position within a genome, have been reported in the evolutionary and adaptive processes of a variety of organisms [69,70] by introducing new cis-acting factors or renovating transcriptional networks to regulate the expression of genes, including metabolic detoxification-related cytochrome P450 genes in insects [71,72,73,74,75]. By screening the association of ETs with P450 genes in both *Helicoverpa zea* and *D. melanogaster*, Chen and Li [71] reported that several TEs were inserted into 5’ or 3′ flanking regions, the exons or the introns of several xenobiotic-metabolizing cytochrome P450 genes in both insect species, and proposed that TEs were selectively retained within or in close proximity to xenobiotic-metabolizing P450 genes. Chung et al. [72] reported the insertion of a long terminal repeat (LTR) of an Accord retrotransposon in the upstream region of the *Cyp6g1* gene, the gene involved in DDT resistance in *D. melanogaster*. The authors demonstrated that the Accord LTR carried regulatory sequences that increased the expression of Cyp6g1 in detoxification tissues, implying their role in resistance P450 gene regulation and insecticide resistance. A similar result was reported by Schlenke and Begun [73], in which the presence of the insertion of a *Doc* transposable element was correlated with increased abundance of *Cyp6g1* transcription. In the mosquito *Culex quinquefasciatus*, a partial CuRE1 (Culex Repetitive Element 1) transposable element has been found to be inserted directly upstream of *CYP9M10*, a cytochrome P450 gene involved in mosquito resistance to permethrin [74]. A study by Marsano et al. [75] provided an example of a cis-regulatory mechanism through the insertion of TE(s) at the 3’ untranslated region of genes. The above examples provide evidence of TE-mediated innovation in gene regulation. Further characterization of molecular mechanisms involved in this cis- or trans-regulation of resistance P450 genes will be essential in shedding light on the development of P450 detoxification-mediated insecticide resistance in insects. 

Techniques such as in silico homology modeling and the molecular docking of proteins are allowing researchers to construct three-dimensional models of proteins, supporting efforts to develop a deeper understanding of the relationship between the structure of proteins such as P450s and substrates such as insecticides, and thus, provide reasonable explanations for the substrate specificities and metabolic specificities caused by allelic variations or mutations [1,76]. In our study, we employed this technique to further confirm the findings of our transgenic and in vitro metabolism study of CYP6A36, CYP6D10 and CYP4S24. Comparisons of the 3D models for permethrin and the potential PBalc and PBald docking positions among CYP6A36, CYP6A52, CYP6D10 and CYP4S24 not only support the metabolic capacity of these P450s for these substrates, but also indicates that CYP6A36, CYP6A52 and CYP6D10 are likely to produce similar metabolites. In CYP4S24, the low metabolic ability of permethrin observed may be due to the narrow channel opening to the heme iron and the small active site cavity (attributed to the protrusion of two amino acids in the helix I structure), which prevent the access of large molecules such as permethrin into the catalytic hotspot [41,77]. However, this structure does provide a good match for small molecules such as PBalc and PBald. This is probably the main reason why CYP4S24 exhibits the highest metabolic ability toward PBalc and PBald among these three P450s. The differences in metabolic activities among the various P450 enzymes of house flies may well be attributed to structural differences in the P450 proteins that facilitate substrate binding, conduction and activation.

## 4. Materials and Methods

### 4.1. House Fly Strains and Lines

Three house fly strains, five backcross (BC_1_) lines and five homozygous lines were used in this study. ALHF is a wild-type multiple-insecticide-resistant strain collected from a poultry farm in Alabama in 1998 [78]. This strain was further selected with permethrin, a pyrethroid insecticide, for six generations after collection to reach a high level of resistance, and has been maintained under biannual selection with permethrin [25,78]. Aabys is an insecticide-susceptible strain with recessive morphological markers including ali-curve (*ac*), aristapedia (*ar*), brown body (*bwb*), yellow eyes (*ye*) and snipped wings (*snp*) on autosomes 1, 2, 3, 4 and 5, respectively. CS is a wild-type insecticide-susceptible strain that has been maintained in the laboratory for more than five decades. Both the aabys and CS strains were originally obtained from Dr. J. G. Scott (Cornell University).

Five back-cross (BC_1_) lines were generated from a cross of ALHF females and aabys males with the following genotypes: *ac/ac*, +/*ar*, +/*bwb*, +/*ye*, +/*sw* (A_2345_); +/*ac*, *ar/ar*, +/*bwb*, +/*ye*, +/*sw* (A_1345_); +/*ac*, +/*ar*, *bwb/bwb*, +/*ye*, +/*sw* (A_1245_); +/*ac*, +/*ar*, +/*bwb*, *ye/ye*, +/*sw* (A_1235_); and +/*ac*, +/*ar*, +/*bwb*, +/*ye*, *sw/sw* (A_1234_). Five homozygous lines were also generated: *ac*/*ac*, +/+, +/+, +/+, +/+ (A2345); +/+, *ar/ar*, +/+, +/+, +/+ (A1345); (+/+, +/+, *bwb*/*bwb*, +/+, +/+ (A1245); +/+, +/+, +/+, *ye*/*ye*, +/+ (A1235); and +/+, +/+, +/+, +/+, *snp*/*snp* (A1234) [11,25]. The name of each line indicates which of its autosomes bear wild-type markers from ALHF. For instance, the A2345 strain has wild-type markers on autosomes 2, 3, 4 and 5 from ALHF and the mutant marker on autosome 1 from aabys.

### 4.2. Quantitative Real-Time PCR (qRT-PCR)

A total of 20 3-day-old adult female house flies from each of the three house fly strains and house fly lines had their RNA extracted for each experiment using the acidic guanidine thiocyanate-phenol-chloroform method [43]. Total RNA (0.5 µg/sample) from each house fly sample was reverse-transcribed using SuperScript II reverse transcriptase (Stratagene, San Diego, CA, USA) in a total volume of 20 μL. The quantity of cDNAs was measured using a spectrophotometer prior to qRT-PCR, which was performed using the SYBR Green Master Mix Kit and the ABI 7500 Real-Time PCR system (Applied Biosystems, Waltham, MA, USA). Each qRT-PCR reaction (15 μL final volume) contained 1× SYBR Green Master Mix, 1 μL of cDNA, and a P450 gene-specific primer pair (Appendix A) designed according to each of the P450 gene sequences at a final concentration of 0.3–0.5 μM. All sampling, including the ‘no-template’ negative control, was performed in triplicate. The specificity of the PCR reactions was assessed by a melting curve analysis for each PCR reaction using Dissociation Curves software [79]. Relative expression levels for the P450 genes were calculated via the 2^−ΔΔCT^ method using SDS RQ software [80]. The β-actin gene and ribosomal protein S3 (RPS3) were used as endogenous controls, to normalize the expression of the target genes [81]. Each experiment was repeated either three or four times with different preparations of RNA samples. The statistical significance of the gene expressions was calculated using a Student’s *t*-test for all pairwise sample comparisons, and one-way analysis of variance (ANOVA) was utilized for multiple sample comparisons (SPSS 19.0 software); a value of *p* ≤ 0.05 was considered statistically significant. 

### 4.3. Autosomal Linkage Analysis of the P450 Genes in M. domestica

Five back-cross (BC_1_) lines (*ac/ac*, +/*ar*, +/*bwb*, +/*ye*, +/*sw* (A_2345_); +/*ac*, *ar/ar*, +/*bwb*, +/*ye*, +/*sw* (A_1345_); +/*ac*, +/*ar*, *bwb/bwb*, +/*ye*, +/*sw* (A_1245_); +/*ac*, +/*ar*, +/*bwb*, *ye/ye*, +/*sw* (A_1235_); and +/*ac*, +/*ar*, +/*bwb*, +/*ye*, *sw/sw* (A_1234_)) were used in this study. The genotype of each BC_1_ line was homozygous for the recessive mutant allele from aabys and heterozygous for the dominant wild-type alleles from ALHF. Allele-specific PCR was conducted using a two-round PCR strategy as described by Liu and Scott [26]. For the first PCR reaction, allele-independent primer pairs (Appendix A) were used to generate P450 cDNA fragments as follows: the first PCR solution with the cDNA template and a primer pair were heated to 95 °C for 3 min, followed by 35 cycles of 95 °C for 30 s, 60 °C for 30 s and 72 °C for 1 min, then, 72 °C for 10 min. The second PCR reaction utilized 0.5 µL of the first-round PCR reaction solution and the allele-specific primer pair (Appendix A), designed based on the specific sequence of the genes from ALHF by placing a specific nucleotide polymorphism at the 3′ end of the primer to permit preferential amplification of the allele from ALHF. This second PCR reaction sample was heated to 95 °C for 3 min, followed by 35 cycles of 95 °C for 30 s, 62 °C for 30 s and 72 °C for 30 s, then, 72 °C for 10 min. Each experiment was repeated three times with different mRNAs, and the PCR products were sequenced at least once for each gene.

### 4.4. Construction of Transgenic D. melanogaster and Toxicity of Permethrin for the Transgenic Lines 

The full lengths of the candidate up-regulated genes from the ALHF strain of *M. domestica* were amplified from cDNA of *M. domestica* using platinum Taq DNA polymerase High Fidelity (Invitrogen) with specific primer pairs (Appendix A) based on the 5′ and 3′ end sequences of the genes. The PCR products of the full-length genes were purified using a QIAquick Gel Extraction Kit (Qiagen). The purified PCR products were ligated into pCR 2.1 vector using the Original TA Cloning kit (Invitrogen) as described by the manufacturer. The full lengths of the genes were cloned in One Shot TOP10F’ cells using the One Shot TOP10F’ Chemically Competent *E. coli* kit (Invitrogen). Cloning and sequence analyses of the cDNAs were repeated at least three times and three TA clones from each replication were verified via sequencing. The clones were then sub-cloned into a pUASTattB vector (a gift from Dr. Johannes Bischof, University of Zurich). The plasmid of each pUASTattB-up-regulated gene was transformed into the germ line of the M{vas-int.Dm}ZH-2A, M{3×P3-RFP.attP’}ZH-58A strain of *D. melanogaster* (Bloomington stock #24484), resulting in site-specific integration on chromosome 2R (Rainbow Transgenic Flies Inc., Camarillo, CA, USA). These flies were then reciprocally crossed against a W^1118^ strain to obtain transformants with the orange eye phenotype, then, balanced against the *D. melanogaster* balancer strain w[1118]/Dp(1;Y)y[+]; sna[Sco]/CyO, P{ry[+t7.2] = sevRas1.V12}FK1 (Bloomington stock #6312) to generate a homozygous line containing the cytochrome P450 transgene. The insertion of the up-regulated genes in the transgenic fruit fly lines was further confirmed using RT-PCR. The resulting homozygous line was crossed with the GAL4-expressing *D. melanogaster* strain P{Act5C-GAL4}17bFO1 (Bloomington stock #3954), which expresses GAL4 under control of the Act5C promoter, resulting in ubiquitous non-tissue-specific expression. The F1 generation of these crosses expressed GAL4 and contained a single copy of the cytochrome P450 transgene that was under the control of the UAS enhancer. 

Permethrin toxicity bioassays were then conducted 2–3 days post-eclosion of female *Drosophila* of the F1 UAS-GAL4 crosses to examine the toxicity of permethrin to transgenic flies. Briefly, serial concentrations of permethrin solution in acetone ranging from 3 ng/µL to 50 ng/µL, which gave >0 and <100% mortality to the tested insects, were prepared, and 200 µL of each permethrin solution was evenly coated on the inside of individual 20 mL glass scintillation vials. Fifteen female flies were transferred to each of the prepared vials, which were then plugged with cotton balls soaked with 5% sucrose. The vials for the control groups were coated with acetone alone and plugged with identical 5% sucrose-soaked cotton balls. Mortality was scored after 24 h exposure to permethrin. Once the concentration range had been determined, a single concentration (25 ng/µL) that showed a significant difference in mortality between the transgenic and non-transgenic flies was selected for further permethrin bioassays. For each bioassay, a total of four vials each containing 15 2–3-day-old adult female *D. melanogaster* were treated with 200 µL of 25 ng/µL permethrin solution for each strain. Each bioassay was independently replicated three times using only flies that exhibited the correct morphological marker (orange eyes). The *D. melanogaster* strain M{vas-int.Dm}ZH-2A, M{3×P3-RFP.attP’}ZH-58A containing the empty pUAST vector-donated insert, but no transgene from *M. domestica*, was used as the control reference strain. All tests were run at 27 °C and mortality was assessed after 24 h. Bioassay data were pooled and probit analysis was conducted. Significant differences in the resistance levels of the *D. melanogaster* lines were determined based on the non-overlap of 95% confidence intervals. All *D. melanogaster* were reared on Jazz-Mix Drosophila food (Fisher Scientific, Kansas City, MO, USA) at 25 ± 2 °C under a photoperiod of 12:12 (L:D) h.

### 4.5. Recombinant Protein Production

The recombinant P450s and NADPH-cytochrome P450 reductase (CPR) proteins used in this study were produced in *Spodoptera frugiperda* (*Sf*9) insect cells using a BaculoDirect^TM^ baculovirus-mediated expression system (Invitrogen). Briefly, the full-lengths of CYP6A36, CYP6D10, CYP4S24 and NADPH-cytochrome P450 reductase (CPR) were amplified from the house fly ALHF strain with specific primers (Appendix A) and ligated with the BaculoDirect^TM^ Linear DNA via an in vitro Lambda Recombination (LR) reaction. The constructed recombinant baculovirus for each gene was then transfected to *Sf*9 cells using CellfectinR II reagents (Invitrogen) to produce the baculovirus stock solution for further infection. The titers of baculoviruses were later measured via a plaque-forming assay, and a titer of ~2 × 10^8^ pfu/mL P2 virus was used as a final stock to infect cells for the large-scale expression of targeted proteins. According to protocols employed in *Culex* mosquitoes for their P450 characterizations [28,75], the co-expression of house fly P450s and CPR (P450/CPR) in *Sf*9 cells was performed with an optimal multiplicity of infection (MOI) ratio of 10:1, supplemented with 1 μg/mL hemin and 0.1 mM 5-ALA, to achieve the highest P450 and catalytic activities [28,75]. The cell lysate proteins were harvested 72 h post-infection and centrifuged at 1000× *g* for 10 min at 4 °C. The cell pellets were washed twice using ice-cold PBS buffer (pH 7.4), and then, re-suspended in homogenization buffer (0.1 M phosphate (pH 7.4), 1 mM EDTA, 0.5 mM PMSF and 0.25 M sucrose). Subsequently, the dissolved cell lysate was then homogenized using a sonicator and ultracentrifuged at 37,000× *g* for 60 min at 4 °C. The microsomal protein pellets were isolated and dissolved in re-suspension buffer (0.1 M phosphate buffer, pH = 7.4, 1 mM EDTA, 0.1 mM DTT, 1 mM PMSF and 20% glycerol stock) and stored at −80 °C for use in in vitro studies.

### 4.6. P450/CPR Activity Determination

The activities of CYP6A36, CYP6D10 and CYP4S24, with co-expressed CPR as their electron donor, were measured via a 7-ethoxycoumarin *O*-deethylation (ECOD) assay with modifications [28]. Briefly, a mixture of 5 μL microsomal protein of each P450/CPR, 1 μL 7-ethoxycoumarin (50 mM dissolved in acetone), and 85 μL sodium phosphate buffer (0.1 M, pH = 7.5) was added to each well of a 96-well microplate, and 10 μL of NADPH was added to initiate the reaction; then, it was incubated at 30 °C for 30 min with orbital shaking, after which 10 μL oxidized glutathione (30 mM) and 10 μL glutathione reductase (5U) were added to remove the NADPH. After 10 min incubation at room temperature, the reaction was stopped by adding 120 μL of 50% acetonitrile in TRIZMA-base buffer (0.05 M, pH = 10). The amount of 7-hydroxycoumarin (7-OH) released during incubation was quantified using a Cytation 3 imagine reader (BioTekUSA) with 390 nm excitation and 450 nm emission filters. A well containing *Sf*9 cell microsomal protein only, and a well without microsomes, served as controls. A standard curve for 7-hydroxycoumarin (with substrate concentrations ranging from 0.1 to 1.2 mM) was used to calculate the kinetic parameters.

### 4.7. P450/CPR-Mediated In Vitro Metabolism of Permethrin, 3-Phenoxybenzyl Alcohol (PBalc) and 3-Phenoxybenzaldehyde (PBald) 

The method and procedure used for the metabolism experiments were as described in our previous studies [28,82,83]. The in vitro metabolism of permethrin and its metabolites, PBalc and PBald, was conducted by incubating chemicals (20 μM) dissolved in Tris-HCl buffer (0.2 M, pH 7.4)/acetonitrile (1:1, *v*:*v*) with 0.2 μM of different recombinant P450/CPR microsomes and an NADPH-regenerating system (1 mM NADP+, 0.25 mM MgCl_2_) in a total volume of 700 μL; the mixture was incubated at 30 °C with 100 rpm orbital shaking and quenched after 120 min by adding 700 μL ice-cold acetonitrile. Samples were then incubated with shaking for an additional 20 min and centrifuged at 16,000× *g* for 10 min. The control group contained the same components, without NADPH added. The supernatant from each sample was then filtered through 0.45 μm membranes and the amounts of the chemicals were analyzed via HPLC using a Waters 2695 High-Performance Liquid Chromatograph System (Milford, MA, USA) with the conditions employed in previous studies [28]. The permethrin amount was measured as the integrated area under the two peaks of trans-permethrin and cis-permethrin [38], and calculated based on the standard curves of each chemical. Reactions were performed in triplicate and a paired *t*-test of sample reactions (with NADPH) vs. the control (without NADPH) was performed for the statistical measurements of substrate depletion following the method described by Gong et al. [28,75].

### 4.8. In Silico Modeling and Substrate Docking Analysis

In silico 3D structural modeling was performed by the I-TASSER server using a combination of threading and ab initio modeling [84,85,86]. Five models were predicted by I-TASSER for each P450, with the top-scoring model being submitted to the FG-MD server for fragment guided molecular dynamics structure refinement [87]. The model quality was controlled by Ramachandran plots, generated using Procheck (http://services.mbi.ucla.edu/SAVES/; accessed on 15 October 2021) [87] and ProSA-web (https://prosa.services.came.sbg.ac.at/prosa.php; accessed on 15 October 2021) [88,89]. The P450 channels were calculated using CAVER 3.0 (http://caver.cz/index.php; accessed on 15 October 2021) [88,89]. The P450 channels were calculated using CAVER 3.0 (http://caver.cz/index.php; accessed on 15 October 2021) [90,91]; those permitting the passage of a sphere with a maximal radius greater than 1.2 Å were identified, tabulated and named according to the nomenclature proposed by Cojocaru et al. [36]. The volume of the substrate binding cavity was characterized by VOIDOO using a 1.4 Å probe [92], and the proteins and ligands were prepared for docking using Autodock Tools v1.5.6 (http://mgltools.scripps.edu/downloads; accessed on 15 October 2021). Molecular docking was performed using Autodock 4.2. [93]. The ligand permethrin structures were retrieved from the ZINC database [94]. For all dockings, a search space with a grid box of 60 × 60 × 60 Å, centered on the heme iron, was set that corresponded to the substrate recognition sites (SRSs), following those for the CYP2 family proposed by Gohoh [95]. The figures were produced using Pymol (http://www.pymol.org/; accessed on 15 October 2021) [96].

## Figures and Tables

**Figure 1 ijms-24-03170-f001:**
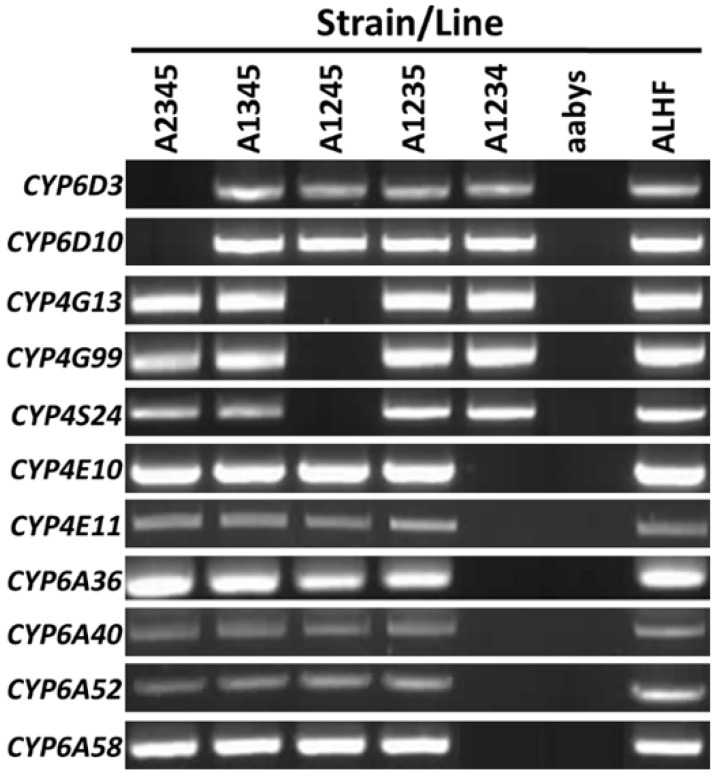
Allele-specific RT-PCR autosomal mapping of *Musca domestica* P450 genes. PCR fragments were generated using an allele-specific primer set according to the sequence for each gene from ALHF. The absence of a PCR product in a house fly line indicates that the gene was located on the corresponding autosome from aabys (i.e., the absence of a band in the A1234 line indicates that the gene was present on autosome 5).

**Figure 2 ijms-24-03170-f002:**
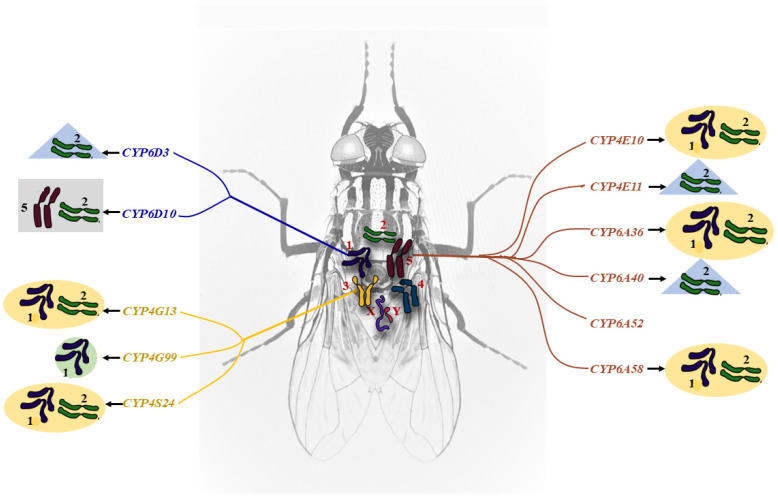
Autosomal interaction in up-regulated P450 gene expression in houseflies. Genes that are located on Autosome 5 are shown in red, on autosome 1 in blue, and on autosome 3 in Yellow. NO P450 genes are identified on either autosome 2 or 4. The autosomes that include factors involved in the interaction and regulation of P450 gene expression are shown with *shaded* highlights.

**Figure 3 ijms-24-03170-f003:**
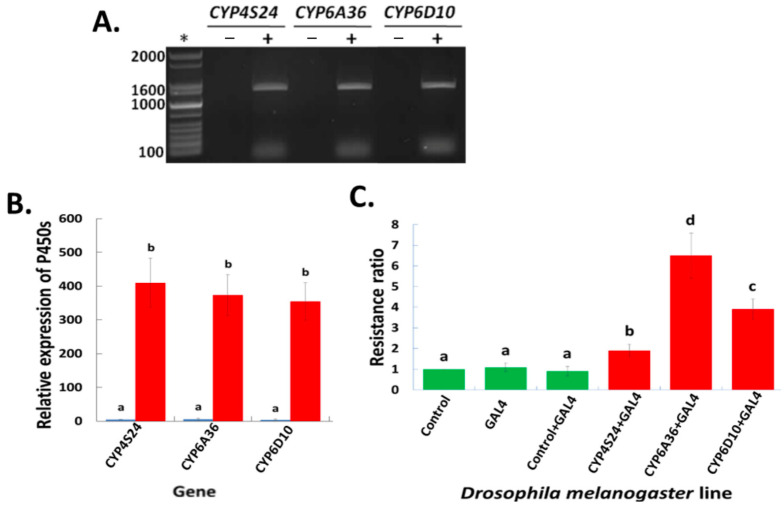
Transgenic study of *CYP4S24*, *CYP6A36* and *CYP6D10* in *D. melanogaster.* (**A**) RT-PCR amplification of the P450 genes in transgenic *D. melanogaster* lines. “−” represents the non-transgenic empty-vector control *D. melanogaster* line. “+” represents the transgenic *D. melanogaster* lines containing house fly P450 genes. * The GelPilot 1Kb (+) ladder was used as a molecular size reference, with the numbers on the figure indicating the DNA band size in bp. (**B**) Transgenic expression of *CYP4S24*, *CYP6A36* and *CYP6D10* in *D. melanogaster*. The relative expressions of the three transgenes were quantified via qRT-PCR. “P450” represents the homozygous transgenic *D.* line with house fly P450 genes, and “P450 × GAL4” represents the F1 generation of the homozygous transgenic *Drosophila* line crossed with GAL4 driver line. Data shown as mean + S.E.M (*n* = 3). (**C**) Toxicity of permethrin to non-transgenic and transgenic *D. melanogaster* lines. Resistance ratios = LD_50_ of *D. melanogaster* lines/LD_50_ of control line. There were no significant differences in expression levels labeled with the same letter (*p* < 0.05).

**Figure 4 ijms-24-03170-f004:**
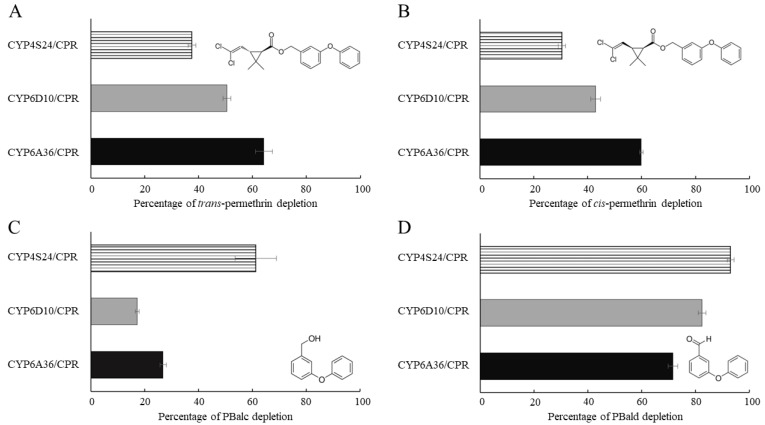
HPLC profiles of permethrin and substrate metabolism mediated by the expressed P450/CPR isomers. Reactions were performed with 20 µM chemical (cis-/trans-permethrin, PBalc or PBald), 1 mM NADPH, 0.25 mM MgCl_2_ and 0.2 μM P450s/CPR microsomal proteins incubated for 2 h at 30 °C. Each P450/CPR protein with NADPH had a control reaction consisting of the same microsome without NADPH. A negative control with Sf9 cell microsomes (0.2 μM) and without P450/CPR expression was used to calculate substrate depletion. (**A**) Percentage of trans-permethrin depletion. (**B**) Percentage of cis-permethrin depletion. (**C**) Percentage of PBalc depletion. (**D**) Percentage of PBald depletion. Data shown as mean ± S.E.M (*n* = 3).

**Figure 5 ijms-24-03170-f005:**
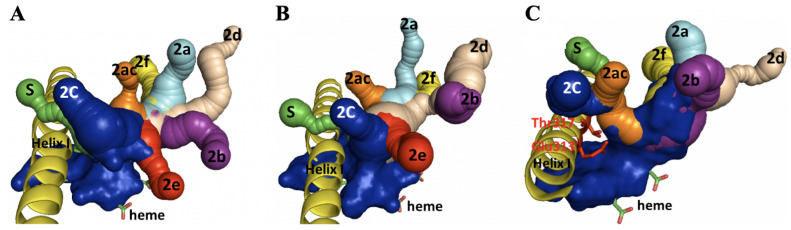
Solvent channel, family 2 channels and active site of P450s. Channels are shown as connected spheres with different colors: cyan (2a), orange (2ac), purple (2b), blue (2c), wheat (2d), red (2e) and yellow (2f). Helix I and heme of P450s are labeled. (**A**) CYP6A36, (**B**) CYP6D10 and (**C**) CYP4S24. Glutamate 313 and Threonine 317 are also labeled.

**Figure 6 ijms-24-03170-f006:**
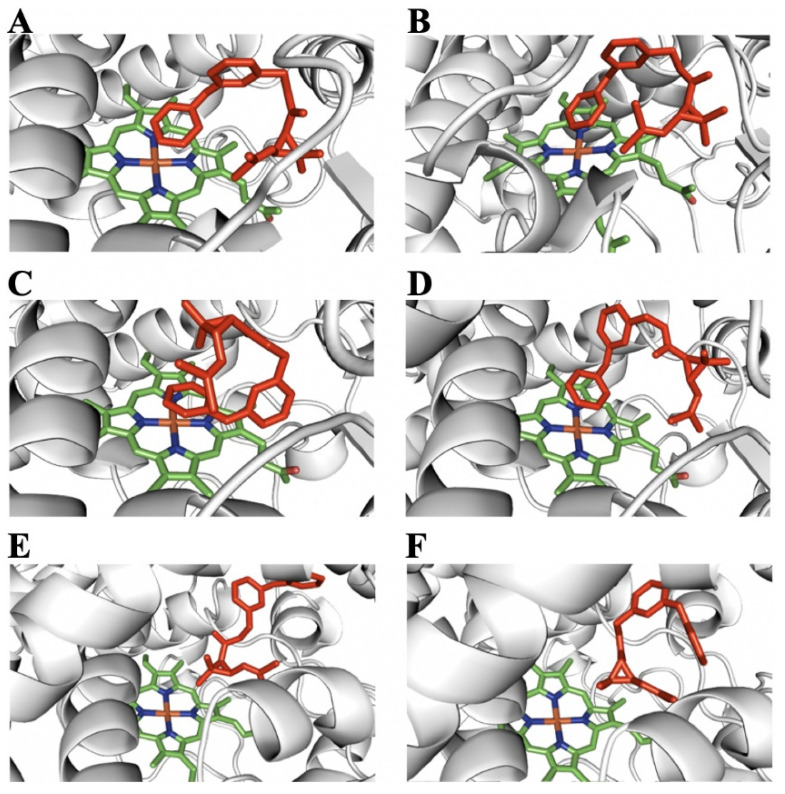
Permethrin binding models for the active site of P450s. The heme group is represented by green sticks and permethrin by red sticks. (**A**) Predicted binding model for cis-permethrin in CYP6A36. (**B**) Predicted binding model for trans-permethrin in CYP6A36. (**C**) Predicted binding model for cis-permethrin in CYP6D10. (**D**) Predicted binding model for trans-permethrin in CYP6D10. (**E**) Predicted binding model for cis-permethrin in CYP4S24. (**F**) Predicted binding model for trans-permethrin in CYP4S24.

**Figure 7 ijms-24-03170-f007:**
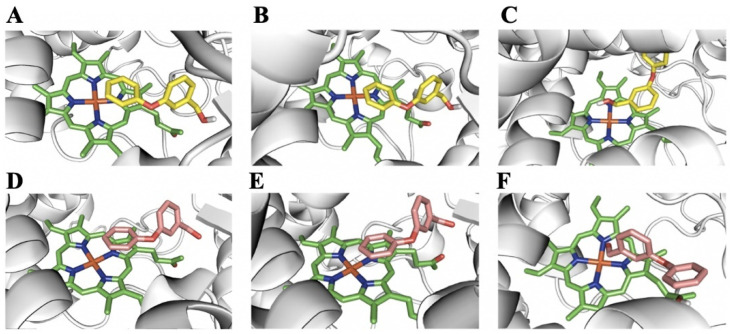
PBalc and PBald binding models for the active site of P450s. The heme group is represented by green sticks, PBalc by yellow sticks and PBald by pink sticks. (**A**) Predicted binding model for PBalc in CYP6A36. (**B**) Predicted binding model for PBalc in CYP6D10. (**C**) Predicted binding model for PBalc in CYP4S24. (**D**) Predicted binding model for PBald in CYP6A36. (**E**) Predicted binding model for PBald in CYP6D10. (**F**) Predicted binding model for PBald in CYP4S24.

**Table 1 ijms-24-03170-t001:** Clan, family and number distribution of selected cytochrome P450 genes in ALHF *Musca domestica*.

CYP Clan	CYP Family *	Number of CYP Members
Clan 2	CYP18	1
	CYP304	1
	CYP305	1
	CYP306	1
Clan 3	CYP6	27
	CYP9	6
	CYP28	3
	CYP310	1
	CYP317	1
	CYP438A4	1
Clan 4	CYP4	23
	CYP311	1
	CYP3073	3
	CYP313	2
	CYP318	1
Mitochondrial CYP	CYP12	9
	CYP301	1
	CYP302	1
	CYP314	1
	CYP315	1

* The P450 gene sequence information was generated from the RNA-seq (NCBI:SRR521288, NCBI:SRR521289) and genome (PRJNA210139, PRJNA176013) analysis of *Musca domestica*.

**Table 2 ijms-24-03170-t002:** Relative expression levels and the predicted autosomal interactions of the up-regulated P450 genes among ALHF strain and other lines of *M. domestica*.

	Relative Gene Expression in Strains/Lines ^§^					
Gene	ALHF	A2345	A1345	A1245	A1235	A1234
*CYP4E10*	274.42 ± 22.40	125.67 ± 10.39 *	144.62 ± 12.62 *	257.81 ± 23.61	268.61 ± 25.8	66.6 ± 6.74 *
*CYP4E11*	5.86 ± 0.55	6.31 ± 0.48	2.61 ± 0.17 *	5.99 ± 0.71	5.80 ± 0.36	2.71 ± 0.26 *
*CYP4G13*	2.13 ± 0.13	1.04 ± 0.10 *	1.22 ± 0.09 *	1.04 ± 0.04 *	2.01 ± 0.12	1.91 ± 0.08
*CYP4G99*	4.52 ± 0.31	2.12 ± 0.22 *	4.28 ± 0.41	1.38 ± 0.13 *	4.43 ± 0.45	4.43 ± 0.23
*CYP4S24*	2.94 ± 0.08	1.03 ± 0.05 *	1.14 ± 0.06 *	0.83 ± 0.03 *	2.83 ± 0.49	3.04 ± 0.53
*CYP6A36*	7.03 ± 0.64	2.60 ± 0.21 *	2.80 ± 0.41 *	6.63 ± 0.62	7.01 ± 1.04	1.72 ± 0.19 *
*CYP6A40*	2.95 ± 0.22	2.90 ± 0.32	1.16 ± 0.10 *	2.99 ± 0.41	2.68 ± 0.36	1.17 ± 0.05 *
*CYP6A52*	2.74 ± 0.15	2.50 ± 0.22	2.70 ± 0.17	2.54 ± 0.08	3.01 ± 0.24	0.81 ± 0.03 *
*CYP6A58*	4.32 ± 1.02	1.32 ± 0.23 *	1.86 ± 0.36 *	5.63 ± 2.17	4.19 ± 0.22	0.64 ± 0.15 *
*CYP6D3*	2.40 ± 0.13	1.38 ± 0.07 *	1.34 ± 0.06 *	2.25 ± 0.13	2.40 ± 0.37	2.34 ± 0.18
*CYP6D10*	6.24 ± 0.63	2.55 ± 0.37 *	3.06 ± 0.55 *	5.99 ± 0.25	6.19 ± 0.51	2.62 ± 0.32 *

^§^ The relative levels of gene expression are shown as a ratio in comparison with those in aabys flies; the data are shown as the mean ± SEM. * Gene expression value within a given *M. domestica* autosomal line was significantly lower than the expression in the parental ALHF strain at a *p* < 0.05 level of significance.

## Data Availability

The data that support the findings of this study are available from the corresponding author, N.L., upon request.

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
