# Peer review of "Multiple-P450 Gene Co-Up-Regulation in the Development of Permethrin Resistance in the House Fly, Musca domestica"

_ijms, 2023, doi:10.3390/ijms24043170_

Round 1

Reviewer 1 Report

The current study focuses on the molecular mechanism of the evolved resistance of pyrethroids in the house fly, Musca domestica. The findings of this study will greatly contribute to the important function of multiple up-regulated P450 genes in the development of permethrin resistance in M. domestica. In addition, the methods used in the manuscript are scientifically sound, but the results still need to be revised carefully, such as the section of 2.1. Especially, this manuscript has some errors listed in the major and minor concerns which need to be addressed as follow:

Major concerns: 

1. The authors prefer to use the long sentence throughout the manuscript. In fact, the sentence is too long to be understand for the readers what the authors want to express. For example, in Page 2 Lines 58-64 and 64-71. Please pay attention to this issue and revised it properly.

2. In the sections of “4.5.” and “4.6.” of Materials and Methods, I strongly recommended the authors should add other treatments of P450 protein alone, such as  CYP6A36, CYP6D10 and CYP4S24, in order to compare the differences of the metabolic activity of P450 and P450/CPR toward permethrin and its metabolites.

3. As for Table 2, it should be inserted between Line 99 and Line 100. The title needs to be changed into “Relative expression levels and the predicted autosomal interactions of the up-regulated P450 genes among ALHF strain and other lines of M. domestica. The first line of “Relative gene expression” should be revised as “Strain/lines” simply.

4. In the section of “2.1.” in “Results”, the authors should check and correct the number of different P450 genes in Lines 78-83 and 93-99 because it is not consistent with that in Table 1.

5. The full name and its abbreviation should be used correctly. Additionally, the consistency of an abbreviation should be kept throughout the manuscript. For example, P450 or CYP450, CYP/CPR or P450/CPR, P450s/CPR, BC1 or BC1, and so on, the form in bold should be better than others.

Minor concerns:

1. In line 2, Gene should be revised as Genes.

2. In Line 30, D. melanogaster should be corrected as Drosophila melanogaster.

3. In Lines 53-71 and 394-415, please correct the font size of these two paragraphs.

4. In Line 123, plays should be revised as playing.

5. In Lines 155-156, and the other (CYP4G99) by factors on autosome 1. should be corrected as and the other (CYP4G99) by factors on autosomes 1 and 3. 

6. In Line 170, is regulated by factors on autosome 1 and 2. should be corrected as is also regulated by factors on autosomes 1, 2 and 5. 

7. In Lines 173, 185, and 204, Figure 3a, Figure 3b and Figure 3c should be corrected as Figure 3A, Figure 3B and Figure 3C, respectively.

8. In Line 193 and 198, the four transgenes and alphabetic should be revised as the three transgenes and lower-case, respectively.

9. In Lines 207-208, these house fly P450 up-regulated genes should be corrected as these up-regulated P450 genes in house fly.

10. In Line 210, in vitro should be written normally because of italic twice.

11. In Line 246, sf9 should be revised as Sf9 .

12. In Line 257, CYP6A36, CYP6D10 and CYP4S24 should be revised as CYP6A36, CYP6D10 and CYP4S24 with co-expressed CPR as its electron donor.

13. In Line 289, delete E, F, from (Figure 6A, B, C, D, E, F, Table S5).

14. In Lines 307-308, CYP6A36 has the strongest and CYP4S24 the lowest ability to metabolize permethrin should be revised as CYP6A36 and CYP4S24 has the strongest and lowest ability to metabolize permethrin, respectively.

15. In Line 370, the full name of CREM should be used.

16. In Lines 377, 379, 384, the abbreviations of the Latin name for insect species should be used.

17. In Lines 410-411, the second PBalc should be corrected as PBald.

18. In Line 484, please check One Shot TOPO 10F’…”  and One Shot TOP10F’…”, which one is written correctly?

19. In Line 496, Cytochrome P450 transgene should be corrected as cytochrome P450 transgene.

20. In Lines 520, 524, 541, 547, please check the unit and its format of temperature.

21. As for References, please check the style of Refs 20, 29, 85 and 88.

22. As for Supplementary materials, the format of P450 proteins including CYP6A36, CYP6D10 and CYP4S24 in Figure S1 should be written normally, but not in italic. In addition, Table S6 should be corrected as Table S5.

 Finally, I hope the authors can use these to correct the same problem for the rest.

Author Response

Response to the Reviewer #1

The current study focuses on the molecular mechanism of the evolved resistance of pyrethroids in the house fly, Musca domestica. The findings of this study will greatly contribute to the important function of multiple up-regulated P450 genes in the development of permethrin resistance in M. domestica. In addition, the methods used in the manuscript are scientifically sound, but the results still need to be revised carefully, such as the section of “2.1”. Especially, this manuscript has some errors listed in the major and minor concerns which need to be addressed as follow:

Major concerns: 

Comment #1. The authors prefer to use the long sentence throughout the manuscript. In fact, the sentence is too long to be understand for the readers what the authors want to express. For example, in Page 2 Lines 58-64 and 64-71. Please pay attention to this issue and revised it properly.

Answer: Agree. Several of the longer sentences have been rewritten and split where possible.

Comment #2. In the sections of “4.5.” and “4.6.” of Materials and Methods, I strongly recommended the authors should add other treatments of P450 protein alone, such as CYP6A36, CYP6D10 and CYP4S24, in order to compare the differences of the metabolic activity of P450 and P450/CPR toward permethrin and its metabolites.

Answer: Thanks for the Reviewer’s comment. However, by P450 alone, it will have no activity or no function  since to be functionally active, P450s need the cofactor, which is CPR (our lab has also previously tested and approved this for several times). This is the reason why we and other researchers co-expressed P450/CPR for the functional studies. In addition, we are not understanding why we need to compare the differences of the metabolic activity of P450 alone while we already know there is no function of P450 without cofactor.

Comment #3. As for Table 2, it should be inserted between Line 99 and Line 100. The title needs to be changed into “Relative expression levels and the predicted autosomal interactions of the up-regulated P450 genes among ALHF strain and other lines of M. domestica. The first line of “Relative gene expression” should be revised as “Strain/lines” simply.

Answer: Agree. These have been changed.

Comment #4. In the section of “2.1.” in “Results”, the authors should check and correct the number of different P450 genes in Lines 78-83 and 93-99 because it is not consistent with that in Table 1.

Answer: Agree. These have been corrected to be consistent with Table 1.

Comment #5. The full name and its abbreviation should be used correctly. Additionally, the consistency of an abbreviation should be kept throughout the manuscript. For example, P450 or CYP450, CYP/CPR or P450/CPR, P450s/CPR, BC1 or BC1, and so on, the form in bold should be better than others.

 Answer: Agree and thanks for point out. These have been corrected through the text.

Minor concerns:

  1. In line 2, “Gene” should be revised as “Genes”.

Answer: Gene is correct here as it pertains to co-up-regulation, not multiple.

  1. In Line 30, “D. melanogaster” should be corrected as “Drosophila melanogaster”.

Answer: Agree. This has been corrected.

  1. In Lines 53-71 and 394-415, please correct the font size of these two paragraphs.

Answer: Agree. This has been corrected

  1. In Line 123, “plays” should be revised as “playing”.

Answer: Agree. This sentence has been rewritten

  1. In Lines 155-156, “and the other (CYP4G99) by factors on autosome 1.” should be corrected as “and the other (CYP4G99) by factors on autosomes 1 and 3.” 

Answer: In our description, we only consider the trans regulatory factors, which is not in the same autosome in which the P450 gene is locate, thus, here should be “…by trans regulatory factors on autosome 1” since CYP4G99 is located on autosome 3.

  1. In Line 170, “is regulated by factors on autosome 1 and 2.” should be corrected as “is also regulated by factors on autosomes 1, 2 and 5.” 

Answer: Same reasons as the response for concern #5.

  1. In Lines 173, 185, and 204, “Figure 3a”, “Figure 3b” and “Figure 3c” should be corrected as “Figure 3A”, “Figure 3B” and “Figure 3C”, respectively.

Answer: Agree. This has been corrected.

  1. In Line 193 and 198, “the four transgenes” and “alphabetic” should be revised as “the three transgenes” and “lower-case”, respectively.

Answer: Agree. These have been corrected.

  1. In Lines 207-208, “these house fly P450 up-regulated genes” should be corrected as “these up-regulated P450 genes in house fly”.

Answer: Agree. These have been corrected.

  1. In Line 210, “in vitro” should be written normally because of italic twice.

Answer: This has been switched to boldface for the header.

  1. In Line 246, “sf9” should be revised as “Sf9” .

Answer: This has been corrected.

  1. In Line 257, “CYP6A36, CYP6D10 and CYP4S24” should be revised as “CYP6A36, CYP6D10 and CYP4S24 with co-expressed CPR as its electron donor”.

Answer: This has been added.

  1. In Line 289, delete “E, F,” from “(Figure 6A, B, C, D, E, F, Table S5)”.

Answer: These have been deleted.

  1. In Lines 307-308, “CYP6A36 has the strongest and CYP4S24 the lowest ability to metabolize permethrin” should be revised as “CYP6A36 and CYP4S24 has the strongest and lowest ability to metabolize permethrin, respectively”.

Answer: This sentence has been rewritten.

  1. In Line 370, the full name of CREM should be used.

Answer: This has been corrected.

  1. In Lines 377, 379, 384, the abbreviations of the Latin name for insect species should be used.

Answer: These have been corrected.

  1. In Lines 410-411, the second “PBalc” should be corrected as “PBald”.

Answer: This has been corrected.

  1. In Line 484, please check “One Shot TOPO 10F’…”  and “One Shot TOP10F’…”, which one is written correctly?

Answer: This has been corrected.

  1. In Line 496, “Cytochrome P450 transgene” should be corrected as “cytochrome P450 transgene”.

Answer: This has been corrected throughout.

  1. In Lines 520, 524, 541, 547, please check the unit and its format of temperature.

Answer: These have been corrected to be consistent.

  1. As for References, please check the style of Refs 20, 29, 85 and 88.

Answer: These have been corrected.

  1. As for Supplementary materials, the format of P450 proteins including CYP6A36, CYP6D10 and CYP4S24 in Figure S1 should be written normally, but not in italic. In addition, Table S6 should be corrected as Table S5.

 Finally, I hope the authors can use these to correct the same problem for the rest.

            Answer: Agree. All these have been corrected.

            Finally, we would express out thanks to the reviewer for the thoroughly review of the manuscript and valuable and thoughtful comments.

Reviewer 2 Report

The manuscript is well written and the conclusions are supported by the results described.

It would be nice to see in the discussion (or even in the introduction) that part of the known variability that causes insecticide resistance in insects in due to the activity of transposasble elements. Transposons can easily introduce new cis acting factors that can rewire transcriptional networks, and this is known in mammals (10.1098/rstb.2019.0347) and insect (0.3390/biology9020025). This phenomenon can induce insecticide resistance as demonstrated in Drosophila (10.1016/j.gene.2005.06.00; 10.1534/genetics.106.066597; 10.1073/pnas.0303793101) and in other insects (10.1186/1471-2148-7-46; 10.1016/j.ibmb.2012.06.003). 

I suggest including a discussion on this topic that seems to be relevant and can improve the impact of the manuscript.

Author Response

Response to the Reviewer #2

Comments: The manuscript is well written and the conclusions are supported by the results described.

It would be nice to see in the discussion (or even in the introduction) that part of the known variability that causes insecticide resistance in insects in due to the activity of transposasble elements. Transposons can easily introduce new cis acting factors that can rewire transcriptional networks, and this is known in mammals (10.1098/rstb.2019.0347) and insect (0.3390/biology9020025). This phenomenon can induce insecticide resistance as demonstrated in Drosophila (10.1016/j.gene.2005.06.00; 10.1534/genetics.106.066597; 10.1073/pnas.0303793101) and in other insects (10.1186/1471-2148-7-46; 10.1016/j.ibmb.2012.06.003). 

I suggest including a discussion on this topic that seems to be relevant and can improve the impact of the manuscript.

Answer: First, thanks for the review’s positive comments on the manuscript. Agree with the reviewer. Although our study mainly focuses on the identification of key P450s with activity to metabolize permethrin and genome rearrangement and disruption from transposable elements was not the focus of our work, however, as reviewer mentioned that discussion of this topic might be interested to readers. Thus, we have provided a paragraph on this topic in the discussion section.

Round 2

Reviewer 1 Report

The authors have revised the manuscript according to the reviewer's suggestions and answered or explained the questions point by point. Im satisfied with almost all revisions they have made, except for their responses to the 2nd question of Major Concerns and the 5th and 6th questions of Minor Concerns. I suggest the authors should need to improve them in the revised manuscript.

Major concerns: 

As for the 2nd question, I suggest the authors should determine the content of recombinant P450s in microsomal protein by reduced CO-difference spectra assay to provide valuable information of the co-expression of P450/CPR in the section of 4.5 Recombinant protein production as previously reported by the team of Wu et al. (2018, 2022) .

References:

1. Wang, H., Shi, Y., Wang, L., Liu, S., Wu, S., Yang, Y., Feyereisen, R., Wu, Y. CYP6AE gene cluster knockout in Helicoverpa armigera reveals role in detoxification of phytochemicals and insecticides. Nat. Commun. 2018, 9, 4820, doi.org/10.1038/s41467-018-07226-6.

2. Shi, Y., Sun, S., Zhang, Y., He, Y., Du, M., ÓReilly, A.O., Wu, S., Yang, Y., Wu, Y. Single amino acid variations drive functional divergence of cytochrome P450s in Helicoverpa species. Insect Biochem. Mol. Biol. 2022, 146, 103796.

 Minor concerns:

As for the 5th and 6th questions, I did not find the trans regulatory factors the authors want to explain in the original manuscript. Some errors can be observed in Lines 158-161 in the revised manuscript after their revision. According to their explanation, the sentence should be correcteded as “the expression of CYP4S24, located on autosome 3 is regulated by trans regulatory factors on autosomes 1 and 2; the expression of CYP6A36, located on autosome 5, is regulated by trans regulatory factors on autosomes 1 and 2; and the expression of CYP6D10, located on autosome 1, is regulated by trans regulatory factors on autosomes 2 and 5.

In addition, there are still some format issues to be addressed. For example, the number of autosome should be replaced the Roman numerals with Arabic numerals, and the P450 gene should be written in italic in the Figure 2; “20µM chemical (cis/trans permethrin, PBalc, or PBald), 1mM”, “Sf9 cell microsomes (0.2 uM)” should be corrected in the legend of Figure 4. Therefore, I really hope the authors correct the same problem carefully throughout the revised manuscript.

Author Response

Reviewer #1

The authors have revised the manuscript according to the reviewer's suggestions and answered or explained the questions point by point. I’m satisfied with almost all revisions they have made, except for their responses to the 2nd question of Major Concerns and the 5th and 6th questions of Minor Concerns. I suggest the authors should need to improve them in the revised manuscript.

Answer: Thanks the reviewer’s positive comments on the revision.

Major concerns: 

Comments: As for the 2nd question, I suggest the authors should determine the content of recombinant P450s in microsomal protein by reduced CO-difference spectra assay to provide valuable information of the co-expression of P450/CPR in the section of “4.5 Recombinant protein production” as previously reported by the team of Wu et al. (2018, 2022) .

References:

  1. Wang, H., Shi, Y., Wang, L., Liu, S., Wu, S., Yang, Y., Feyereisen, R., Wu, Y. CYP6AE gene cluster knockout in Helicoverpa armigerareveals role in detoxification of phytochemicals and insecticides. Nat. Commun. 2018, 9, 4820, doi.org/10.1038/s41467-018-07226-6.
  2. Shi, Y., Sun, S., Zhang, Y., He, Y., Du, M., ÓReilly, A.O., Wu, S., Yang, Y., Wu, Y. Single amino acid variations drive functional divergence of cytochrome P450s in Helicoverpa species. Insect Biochem. Mol. Biol.2022, 146, 103796.

Answer: Thanks for sharing these two excellent papers with us. We agree that CO-difference spectra assay is excellent for visualize P450 spectrum for expression.  However, instead of using CO-difference spectra assay, we have used a very conventional P450 specific method ECOD, which has been accepted/approved by many researchers, in our study to prove that P450 was expressed in Sf9 cell. EDOD activity is not only prove the expression of the P450 genes in the cells, but also verify the functional activity of the expressed P450 in the cells.

Using either CO-difference spectra assay or P450 specific activity assay, there is not impact on our data analysis, P450 metabolism results, and conclusion of our current finding.

However, we gained information from these two shared papers for supporting the multiple P450 genes involved in insecticide resistance as well as the mutations of P450s play the important role in the P450-mediated resistance besides the up-regulation of the P450 genes. We have cited these studies in our discussion section.

Minor concerns:

Comments: As for the 5th and 6th questions, I did not find the trans regulatory factors the authors want to explain in the original manuscript. Some errors can be observed in Lines 158-161 in the revised manuscript after their revision. According to their explanation, the sentence should be correcteded as “the expression of CYP4S24, located on autosome 3 is regulated by trans regulatory factors on autosomes 1 and 2; the expression of CYP6A36, located on autosome 5, is regulated by trans regulatory factors on autosomes 1 and 2; and the expression of CYP6D10, located on autosome 1, is regulated by trans regulatory factors on autosomes 2 and 5.

Answer: Agree, the changes have been made.

Comments: In addition, there are still some format issues to be addressed. For example, the number of autosome should be replaced the Roman numerals with Arabic numerals, and the P450 gene should be written in italic in the Figure 2; “20µM chemical (cis/trans permethrin, PBalc, or PBald), 1mM”, “Sf9 cell microsomes (0.2 uM)” should be corrected in the legend of Figure 4. Therefore, I really hope the authors correct the same problem carefully throughout the revised manuscript.

Answer: Agree, Changes have made accordingly.

Reviewer 2 Report

I thank the Authors for acknowledging the role of transposable elements in generating resistance-related phenotypes. 

I think that both the suggested reviews (10.1098/rstb.2019.0347 and 10.3390/biology9020025) are worth of mentioning to introduce the role of TEs in the evolutionary and adaptive processes in a variety of organisms. Similarly, the work from Marsano et al (10.1016/j.gene.2005.06.005) provides an example of cis-contribution when insertion occurs at the 3' of genes. The latter example gives the readers an idea of the great potential of transposition-mediated innovation in gene regulation.

Author Response

Comments: I thank the Authors for acknowledging the role of transposable elements in generating resistance-related phenotypes. 

I think that both the suggested reviews (10.1098/rstb.2019.0347 and 10.3390/biology9020025) are worth of mentioning to introduce the role of TEs in the evolutionary and adaptive processes in a variety of organisms. Similarly, the work from Marsano et al (10.1016/j.gene.2005.06.005) provides an example of cis-contribution when insertion occurs at the 3' of genes. The latter example gives the readers an idea of the great potential of transposition-mediated innovation in gene regulation.

Answer. Agree. These three papers have been introduced in the discussion section.